# Deficiency of *ValRS-m* Causes Male Infertility in *Drosophila melanogaster*

**DOI:** 10.3390/ijms25137489

**Published:** 2024-07-08

**Authors:** Xin Duan, Haolin Wang, Zhixian Cao, Na Su, Yufeng Wang, Ya Zheng

**Affiliations:** 1School of Life Sciences, Central China Normal University, Wuhan 430079, China; duanxin@mails.ccnu.edu.cn (X.D.); wanghl1009@163.com (H.W.); caozxsweet722@163.com (Z.C.); yfengw@ccnu.edu.cn (Y.W.); 2School of Life Sciences, Shanghai Normal University, Shanghai 200234, China; sn2751048220@163.com

**Keywords:** *ValRS-m*, *Drosophila melanogaster*, mitochondria, spermatogenesis, RNA-seq

## Abstract

*Drosophila* spermatogenesis involves the renewal of germline stem cells, meiosis of spermatocytes, and morphological transformation of spermatids into mature sperm. We previously demonstrated that *Ocnus* (*ocn*) plays an essential role in spermatogenesis. The *ValRS-m* (*Valyl-tRNA synthetase*, *mitochondrial*) gene was down-regulated in *ocn* RNAi testes. Here, we found that *ValRS-m*-knockdown induced complete sterility in male flies. The depletion of *ValRS-m* blocked mitochondrial behavior and ATP synthesis, thus inhibiting the transition from spermatogonia to spermatocytes, and eventually, inducing the accumulation of spermatogonia during spermatogenesis. To understand the intrinsic reason for this, we further conducted transcriptome-sequencing analysis for control and *ValRS-m*-knockdown testes. The differentially expressed genes (DEGs) between these two groups were selected with a fold change of ≥2 or ≤1/2. Compared with the control group, 4725 genes were down-regulated (dDEGs) and 2985 genes were up-regulated (uDEGs) in the *ValRS-m* RNAi group. The dDEGs were mainly concentrated in the glycolytic pathway and pyruvate metabolic pathway, and the uDEGs were primarily related to ribosomal biogenesis. A total of 28 DEGs associated with mitochondria and 6 meiosis-related genes were verified to be suppressed when *ValRS-m* was deficient. Overall, these results suggest that *ValRS-m* plays a wide and vital role in mitochondrial behavior and spermatogonia differentiation in *Drosophila*.

## 1. Introduction

Recent studies have shown that up to 10–15% of human couples in the world are currently infertile, and about 30–40% of these cases are due to male infertility factors, including asthenospermia and oligospermia [1]. Multiple genetic manipulation tools in *Drosophila melanogaster* provide a significant opportunity to study the conserved signaling pathways during spermatogenesis [2]. The spermatogenesis of *Drosophila* mainly includes germline stem cell renewal, primary spermatocyte meiosis, and spermatid-to-sperm morphological transformation [3]. In *Drosophila* testes, germ stem cells (GSCs) undergo asymmetric divisions, producing new GSCs and gonialblasts (GBs). Next, one GB undergoes exactly four rounds of mitotic transit-amplifying (TA) divisions to generate a cluster of 16 interconnected spermatogonia, which differentiate to become spermatocytes and then undergo meiosis, resulting in a cluster of 64 spermatids. The spermatids in the cyst undergo lengthening and individualization, eventually becoming fully mature sperm cells [4].

The balance between animal germ cell proliferation and differentiation is critical for germline homeostasis. It is tightly regulated by the stem cell niche and spermatogonia transit-amplification (TA) division [5]. Germline differentiation must be coordinated with GSC self-renewal, and spermatocytes must undergo meiosis at the right time [6]. Previous studies have shown that transcriptional activation is the primary mechanism of the mitosis to meiosis transition, and the expression of multiple genes changes during the transition from S16 to primary spermatocytes [7]. The activation of tMAC (testis-specific meiotic arrest complex) promotes spermatocyte meiosis [8,9]. The mutation of the *aly* gene, as one of the tMAC members, can result in the accumulation of spermatocytes in the testes and abnormal spermatocyte division, thus eventually causing male infertility [10].

Recent studies have shown significant mitochondrial clustering during TA transformation in *Drosophila* spermatogenesis [11]. In the primary spermatocyte before meiosis, the spherical mitochondria are clustered near the nucleus. After meiosis II is completed, hundreds of mitochondria fuse into two long mitochondrial derivatives, which intertwine to form a spherical aggregate near the nucleus, known as the accessory nucleus (nebenkern). During the elongation of sperm cells, the behavior of mitochondrial derivatives also changes. One of them becomes a major mitochondrial derivative, with paracrystalline accumulation. The other one becomes the minor mitochondrial derivative, which reduces in size and volume [12]. The *Fzo* (fuzzy onions) gene was discovered to regulate mitochondrial fusion in *Drosophila*. It belongs to one member of the evolutionally conserved mitochondrial fusion protein family (Mfns), encoding a sizeable transmembrane GTPase associated with dynamin, which mediates mitochondrial fusion during spermatogenesis. Deletion of the *Fzo* gene causes the mitochondria to fail to fuse properly, resulting in the malformation of the nebenkern and male sterility [13].

The human gene *VARS2* (*Valyl-tRNA synthetase 2*) is related to *COXPD20* (*combined oxidative phosphorylation deficiency 20*) [14]. Studies have shown that *VARS2* mutation can cause autosomal recessive mitochondrial encephalomyopathy [15,16]. The *VARS2* mutations can also lead to varying degrees of developmental delay, axial hypotonia, limb spasms, and even premature death [17]. The homolog of *VARS* is the *ValRS-m* gene of flies, which acts exclusively in the mitochondria and is highly expressed in the testis. Notably, we previously found that the knockdown of *ocnus*, which encodes histidine phosphatase in the fly testis, can lead to abnormal development of the testis and male sterility [18]. Through comparative proteomics detection, the *ValRS-m* protein was significantly down-regulated in *ocn*-knockdown fly testes. However, its specific function in *Drosophila* has not been reported so far.

In the present study, we explored the role of *ValRS-m* in *Drosophila* spermatogenesis. *ValRS-m* knockdown in the testes caused male sterility and early spermatogenesis defects in flies. The *ValRS-m*-knockdown testes exhibited severe defects in the transition process from spermatogonia to spermatocytes. Immunofluorescence staining and electron microscopic examination revealed that *ValRS-m*-knockdown disrupts mitochondrial fusion and ATP synthesis during spermatogenesis. In addition, a total of 7710 genes were identified as the DEGs in *ValRS-m*-knockdown testes, including many meiosis-related genes and mitochondrial genes. Together, these findings indicated that *ValRS-m* is required in *Drosophila* spermatogenesis to control mitochondrial behavior and spermatogonia differentiation. Our results not only further elucidate the regulatory mechanism of animal reproductive development but also provide a valuable theoretical reference for the further study of human spermatogenesis and mitochondrial diseases.

## 2. Results

### 2.1. Knockdown of ValRS-m in Testes Caused Male Infertility

Our previous studies have shown that *ocn* knockdown leads to complete sterility in male flies, and the expression of ValRS-m is significantly reduced in the *ocn* RNAi testes. Therefore, to reveal the function of *ValRS-m* in *Drosophila* male fertility, we detected the expression level of the *ValRS-m* gene in the testes and ovaries of 1-day-old adult flies using quantitative RT-PCR. As shown in Figure 1A, the transcript of *ValRS-m* was considerably higher in the testis than in the ovary. To further examine the role of the *ValRS-m* in male fertility, the UAS-Gal4 system was adopted to manipulate the expression levels of *ValRS-m* in the fly testis. It is known that bag-of-marbles (bam) is expressed in spermatogonia as a critical regulator of spermatogonia proliferation and becomes silenced in spermatocytes [19]. By crossing UAS-*ValRS* RNAi male flies with bam-gal4 females, *ValRS-m* was knocked down specifically in the spermatogonia of the male offspring (bamGal4 > *ValRS-m*-hp). The knockdown efficiency in relation to *ValRS-m* expression in these testes was validated via qRT-PCR (Figure 1B). To assess whether *ValRS-m*-knockdown in fly testes may affect male fertility, we crossed the 1-day-old *ValRS-m* RNAi males with 3-day-old w1118 virgin females and collected the eggs. Surprisingly, in three independent trials, the hatching rates of the eggs in groups crossed with *ValRS-m* RNAi males were zero. In contrast, 89.57 ± 0.35% of eggs from the crosses with control males hatched (Figure 1C).

The *Drosophila* testis is a blunt-ended, coiled tube that supports the production of sperm throughout the life of the fly (Figure 2A). We dissected these adult testes and checked the spermatogenesis progress in these flies. DNA staining can visualize the early-stage nuclei at the apex of the testis and the nuclei of clusters of elongated spermatozoa at the tail of the testis [20]. We found that testes from *ValRS-m*-knockdown males exhibited a smaller testicular size compared with the control group (Figure 2B,C). Secondly, we observed that in the control testes, the spermatid nuclei bundles were cone-shaped and the sperm nuclei were concentrated; however, navicular nuclei and spermatid nuclei bundles had disappeared in the *ValRS-m*-knockdown testes (Figure 2C′). Finally, numerous mature sperm were stored in the seminal vesicles of the control testes (Figure 2B″). In contrast, no mature sperm were found in the seminal vesicles of the *ValRS-m* RNAi testes (Figure 2C″). Taken together, these results suggest that *ValRS-m* is essential for sperm production and male fertility.

### 2.2. ValRS-m-Knockdown Testes Accumulated Spermatogonia

To explore the underlying cause of the male sterility induced by *ValRS-m*-knockdown in fly testes, the *bam gvp >+* and *bam Gal4 > ValRS-m-hp* fly testes were dissected and stained with Vasa, DAPI, and spectrin to see whether spermatogenesis was disrupted. During *Drosophila* spermatogenesis, each GSC-derived gonialblast cell first undergoes four rounds of mitotic divisions to generate 16 spermatogonia, which then differentiate into spermatocytes and enter meiosis to produce spermatids. Here, the germ-cell marker Vasa antibody staining showed that the small early-stage germline cells and bigger spermatocytes could be seen in control testes (Figure 3A,A′). However, *ValRS-m*-knockdown testes accumulated numerous spermatogonia-like cells in most area of the testes (Figure 3B,B′).

Germline cells develop within a cyst, where they are interconnected by cytoplasmic bridges. The fusome (labeled by the anti-spectrin antibody), an organelle resembling the endoplasmic reticulum that is specific to the germline, exhibits a punctate morphology in GSCs and GBs, thin-branched fusomes in spermatogonia, and thick-branched fusomes in spermatocytes [21]. In the control testes, there were round fusomes (blue arrows) and thin-branched fusomes (yellow arrows) in spermatogonia and thick-branched fusomes (white arrows) in spermatocytes at the apex of the testes (Figure 3A1). However, compared with the control group, the *ValRS-m*-knockdown testes accumulated abundant punctate and thin-branched fusomes. The thick-branched fusomes could not be seen in *ValRS-m*-knockdown testes (Figure 3B1). These staining results suggested that the knockdown of *ValRS-m* may affect spermatogonia differentiation and TA division in *Drosophila*.

### 2.3. ValRS-m-Knockdown Disrupts Mitochondrial Behavior during Spermiogenesis

Mitochondrial fusion is necessary for normal spermatogenesis in *Drosophila* [22]. Therefore, to further explore the function of *ValRS-m* during germline differentiation, we used mitochondrial staining (Mito) to label the functional mitochondria and anti-VDAC1/Porin antibody to label the outer mitochondrial membrane by immunofluorescence staining. The results show that compared with the control testes, the mitochondrial content in the *ValRS-m*-knockdown testes was significantly increased, and most of the locations were concentrated at the posterior segment of the testes, indicating a significant increase in cell proliferation (Figure 4A′,D). Moreover, the VDAC1 morphology in the control testes gradually aggregated from granular to linear (Figure 4B). In contrast, in the *ValRS-m*-knockdown testes, only diffuse signals were observed, and no linear mitochondria were found (Figure 4B′). It is known that the morphology of mitochondria is correlated with their metabolic activity [23]. The inner mitochondrial membrane morphology and cristae organization are crucial for the assembly and proper function of the oxidative phosphorylation (OXPHOS) system, which is responsible for ATP synthesis [24]. ATP5A (also named Blw) is the alpha subunit of the mitochondrial F1F0 ATP synthase complex V, which participates in the oxidative phosphorylation pathway. Compared with the control group, the ATP5A signal was weakened at the apical tips of *ValRS-m*-knockdown testes (Figure 4C′).

Studies have revealed that abnormal mitochondrial morphology is coupled with failed spermatid development, such as in *Cnt1* mutation and *Cyt-c1L* RNAi testes [25,26]. Thus, using electron microscopy, we further examined the structure of the elongating spermatids. In the control elongating spermatids, we observed axoneme structures with nine outer pairs and a central pair of doublet microtubules (Figure 4F,G). In the *ValRS-m*-knockdown testes, no axoneme was observed (Figure 4H). In addition, each elongating spermatid in the normal testes contained one mitochondrial derivative coupled with one axoneme (Figure 4G,E). However, in the *ValRS-m*-knockdown testes, the mitochondrial derivatives became fewer and looser (Figure 4H). Taken together, our data demonstrated that the knockdown of *ValRS-m* disrupted the normal mitochondrial behavior, which might result in an energy deficit to block sperm production.

### 2.4. Knockdown of ValRS-m Induces Apoptosis in Testes

Considering that the knockdown *of ValRS-m* caused spermatid nuclei bundle and mature sperm disappearance in the seminal vesicle, we speculated that cell death could be induced. Using the TUNEL assays, we observed that the apoptotic signals in the apical region of the *ValRS-m*-knockdown and control testes were similar; however, much more apoptotic signals were detected at the mid-posterior region of the *ValRS-m* RNAi testes (Figure 5B″,C) than that in the normal testes (Figure 5A″). Notably, most of the apoptotic signals appeared spherical, suggesting that apoptosis may occur in the whole cyst of spermatogonia that failed to go through further processes.

### 2.5. Absence of ValRS-m Alters the Expression Profiles of Genes in Testes

To evaluate the effect of *ValRS-m* on the expression of spermatogenesis-related genes, we performed transcriptomic expression profiling of the testes in response to *ValRS-m* depletion. A total of 17,143 genes were identified through this sequencing. The numbers of expressed genes, total mapped reads, and unique matches for each sample are shown in Appendix A. A total of 7710 differentially expressed genes (DEGs) were identified in the *ValRS-m*-knockdown testes, including 2985 up-regulated and 4725 down-regulated DEGs (the absolute value of log2 fold-change > 1 and *p* value < 0.05) (Figure 6A). Among the DEGs, the down-regulated genes were mainly highly expressed in the testes, and the up-regulated genes were mainly highly expressed in the ovaries (Figure 6B), implying that *ValRS-m* plays a critical role in *Drosophila* reproduction.

To further analyze the biological events induced by *ValRS-m* RNAi during spermatogenesis, we performed gene ontology (GO) enrichment analysis for these DEGs according to their functional categorization. The up-regulated DEGs were primarily enriched in processes including cytoplasmic translation (87 genes), rRNA processing (63 genes), and the cytosolic large ribosomal subunit (49 genes) (Figure 6C), while the down-regulated DEGs were enriched in processes such as microtubule-based movement (71 genes), microtubule-associated complex (64 genes), and the ciliary part (53), and 46 DEGs were involved in mitochondrial transport (Figure 6D). Moreover, we performed KEGG pathway analysis for these DEGs. Among the up-regulated genes, the top three KEGG pathways with the highest enrichment were the ribosome, Huntington’s disease, and spliceosome (Figure 6E). Meanwhile, the down-regulated DEGs were mainly enriched in glycolysis/gluconeogenesis, the glucagon signaling pathway, and the pyruvate metabolism pathway (Figure 6F).

To confirm these effects, further validation of the candidate genes using qRT-PCR was conducted, and the primer sequences are shown in Appendix A. We detected the expression of 28 DEGs associated with mitochondria between the *ValRS-m*-knockdown and control testes (Table 1). In line with the transcriptome results, their expression was significantly decreased in the *ValRS-m*-knockdown testes (Figure 7A). Referring to the FlyBase database, the DEGs related to mitochondrial morphology are *mics1*, *Mul1*, and *Knon*. The DEGs related to mitochondrial electron transport are *UQCR-11L*, *Cyt-c-d*, and *ND-B14.5AL*. Furthermore, five genes (*CysRS-m*, *OXA1L*, *VhaM9.7-d*, *Vha100-3*, and *Hsc70-1*) are involved in ATPase-binding activity and ATP synthase. Additionally, GO analysis indicated that six DEGs involved in meiosis were all down-regulated significantly in the *ValRS-m* RNAi testes, which was supported by the qRT-PCR results (Figure 7B). Among these genes, *Cbc* (*crowded by cid*) is essential for meiosis, working together with tRNA splicing endonuclease subunit 54 (*Tsen54*) to regulate the transition to meiosis during spermatogenesis [27]. The other four genes (*Aly*, *Comr*, *Topi*, and *Wuc*) encoding tMAC (testis-specific meiotic arrest complex) members play the role of gene activators in spermatogonia that help to activate meiosis [9]. Taken together, the transcriptional profile analysis and qRT-PCR results demonstrated that *ValRS-m* is required to regulate mitochondrial morphology and function, as well as meiosis-related gene transcription, which will help spermatogonia prepare for their transition toward spermatocytes.

## 3. Discussion

In this study, we observed that the testis from *ValRS-m*-knockdown males was smaller in appearance, lacking spermatocytes, elongated spermatids, and mature sperm in the seminal vesicles. Spermatogenesis is a complex process of cell differentiation, and the transition of the cell state is regulated by a strict cascade [6]. The transition process from spermatogonia to primary spermatocytes is essential in *Drosophila* spermatogenesis, in which large numbers of genes are involved, along with transcriptional and morphological changes, and about 1500 genes are estimated to only be expressed in spermatocytes according to genome-wide microarray data [28,29]. From the RNA-seq results, we have identified that the expression level of the total 7710 genes was significantly changed by *ValRS-m*-knockdown (Appendix A), including the significantly up-regulated genes *bam* and *nup* (spermatogonia-specific) [30,31] and down-regulated genes *Snp* and *Cmet* (spermatocyte-specific) [32], implying the role of *ValRS-m* in regulating the transition from spermatogonia toward spermatocytes in testes, further supported through immunofluorescence staining of Vasa and spectrin (Figure 3).

*ValRS-m* is a nuclear gene encoding for mitochondrial Valyl-tRNA synthetase, which is the powerhouse of eukaryotic cells due to the ability to produce ATP through oxidative phosphorylation [17,33]. ATP production is essential for sperm motility, capacitation, and acrosomal reactions [34]. In somatic cells, as cells enter the interphase, mitochondria exhibit an elongated network pattern and aggregate in the nucleus and pericellular area. In contrast, during most mitosis, mitochondria exhibit a fragmented network pattern scattered throughout the cytoplasm [35,36]. Compared to somatic cells, the changes in germ cells during differentiation and division are more obvious. For example, sperm and eggs undergo meiosis and the complex dynamic distribution of the cytoskeleton. These processes require large amounts of ATP for maintenance [37,38]. In this study, we found that ATP signaling was very powerful in the control testes and highly expressed in both the round and elongated spermatids. However, ATP5A signaling was almost undetectable in the *ValRS-m*-knockdown testes (Figure 4C′). Therefore, these results suggest that the knockdown of *ValRS-m* leads to the obstruction of ATP synthesis during spermatogenesis.

Studies have shown that mitochondria can participate in a series of activities in sperm differentiation, including the fusion and aggregation of early active mitochondria to form a nebenkern [39]. Dorogova et al. (2013) showed that the primary spermatocytes of wild-type *Drosophila* underwent a pre-meiosis growth phase, the number of mitochondria increased significantly, and the mitochondria of spermatocytes were compact and evenly distributed in the cytoplasm [40]. By immunofluorescence staining and TEM, we speculated that the spermatogenesis in *ValRS-m*-knockdown testes may be stagnant in the spermatogonia stage without subsequent meiosis, because the axial and fibrous complex is missing and the nucleus is in a diffuse state in the absence of *ValRS-m*.

This prediction was further confirmed via our subsequent RNA-seq and qRT-PCR results. RNA-seq analysis and qRT-PCR revealed that many meiosis-related genes were significantly decreased in the *ValRS-m* RNAi testes compared to the control testes, including *Cbc* and *Tsen54* (Figure 6B). The expression of the *Cbc* gene encodes a polynucleotide 5′-hydroxyl-kinase required at the transition to meiosis in spermatogenesis, as well as the expression of *Tsen54* being conducive to the transition to meiosis [27]. Furthermore, the primary spermatocytes undergo high levels of transcription in preparation for differentiation into spermatids. A group of genes, when mutated, induce arrest at the primary spermatocyte stage and male sterility. These genes are called tMAC (testis-specific meiotic arrest complex), and their mutants arrest at the G2–M transition of meiosis I. Therefore, tMAC is required for both meiotic cell cycle progression and the onset of spermatid differentiation in spermatocytes [41]. Our qRT-PCR results also showed that the expression levels of the tMAC genes (*Aly*, *Comr*, *Topi*, and *Wuc*) were down-regulated significantly in the absence of *ValRS-m*. Due to *Aly*, *Comr*, and *Topi* only being expressed in primary spermatocytes [10,29,41], the knockdown of *ValRS-m* inhibited spermatocyte production. Moreover, the spermatogonia that did not enter the next step eventually underwent apoptosis, which is consistent with the observed increase in the apoptotic signal tunnel (Figure 5).

Furthermore, we found that *ValRS-m* RNAi interrupts the expression of multiple mitochondrial genes (Figure 7A). These play a critical role in mitochondrial function and spermatogenesis. For example, Cyt-c1, as a subunit of mitochondrial respiratory chain complex III, is involved in oxidative phosphorylation, which mediates electron transport from cytochrome b to cytochrome c [42,43]. The *Sprn* gene, located specifically in the mitochondrial nebenkern, plays a role in sperm elongation [44]. *Mics1* encodes mitochondrial intima proteins and interacts with proteins encoded by *chchd2* to enhance oxidative phosphorylation [45]. Studies have shown that Knon, an unusually large paralogic homologue of ATP synthase subunit d, participates in the internal shaping of the nebenkern, forming mitochondrial membrane during *Drosophila* spermatogenesis. Male flies lacking *knon* are sterile [46]. *Cyt-c1L* encodes a subunit of mitochondrial respiratory chain complex III, which is involved in mitochondrial ATP synthesis and the coupled proton transport process. We previously found that the knockdown of *Cyt-c1L* could result in abnormal spermatocyte apoptosis and immature sperm, resulting in male infertility [26]. Taken together, these studies suggest that the deficiency of *ValRS-m* might inhibit spermatogenesis by affecting the mitochondrial structure and function in flies.

The effect of *ValRS-m* RNAi on *Drosophila* spermatogenesis is extensive and significant. To further reveal its regulatory mechanism in relation to male fly fertility, we have identified 7710 DEGs in *ValRS-m* RNAi testes via RNA-seq, including 4725 down-regulated genes and 2985 up-regulated genes. We found that the metabolic pathways associated with the up-regulated DEGs were most involved in ribosome biogenesis (104 genes), such as RPS3 (ribosomal protein S3), RPL22 (ribosomal protein L22), RPL36 (ribosomal protein L36), etc. Studies have shown that most ribosomal proteins not only play a role in the biological reaction process of ribosomes and protein synthesis but also affect the process of biological somatic activities by performing “extra-ribosomal” functions, which are involved in many biological processes, including growth and development, cell apoptosis, and aging processes [47]. The fruit fly ribosome contains 79 different proteins, among which RPS3 is not only involved in translation and ribosome maturation but can also recognize DNA damage [48]. Importantly, RpS3 plays a critical role in regulating spermatid elongation and individualization [49]. In addition, the deletion of the *RpL22* gene is common in many types of cancer and cell lines [50], and *RPL36* knockdown reduced the number of germ cells in the testes and induced few or no mature sperm in the seminal vesicles [51].

Among the DEGs in *ValRS-m* RNAi testes, we identified 228 DEGs that have been reported to be associated with spermatogenesis according to GO analysis. Among these DEGs, S-Lap (Sperm-Leucylaminopeptidase) proteins are crucial constituents of the paracrystalline material of *Drosophila* sperm, and their mutant alleles could cause male sterility and an abnormal structure of the elongated major mitochondrial derivatives [52]. In addition, the Bb8 (big bubble 8) protein has a crucial role in the normal development and establishment of the mitochondrial derivatives during spermatid elongation [53]. Another gene, *Hsp60C* (*heat shock protein 60C*), is widely expressed and has the activity of hydrolyzing ATP. It is essential for embryonic viability and has important roles in spermatogenesis, possibly through interaction with F-actin in flies [54].

Additionally, the down-regulated DEGs are most enriched in the glycolysis/gluconeogenesis pathway and pyruvate metabolic pathway, and they are also involved in the citric acid cycle and pentose phosphate metabolic pathway; these are all carbohydrate metabolic pathways. Glycolysis is a vital component of semen activity because it is involved in capacitation, providing a second energy source for spermatocyte development. It is closely associated with sperm functional maturation and epididymal transport [55]. Studies have shown that carbohydrate metabolism is closely related to spermatogenesis [56,57]. For example, the testicles promote carbohydrate metabolism in adjacent gut sites through JAK-STAT signaling in fruit flies [10,58]. The male intestine secretes citrate into the adjacent sperm, thus promoting sperm maturation [56]. We have found that the knockdown of *ValRS-m* significantly inhibited the expression of genes involved in carbohydrate synthesis; thus, it may have an impact on the energy required for spermatogenesis. Taken together, it is speculated that the knockdown of *ValRS-m* may affect the function of various ribosomal proteins and carbohydrate metabolism, hence interfering with spermatogenesis.

## 4. Materials and Methods

### 4.1. Fly Stocks

Flies were reared on a standard cornmeal/yeast diet at 25 °C and under non-crowded conditions (200 ± 10 eggs per 50 mL vial of media in a 150 mL conical flask). The transgenic *ValRS-m*-hp *D. melanogaster* line was obtained from the Tsinghua Fly Center (Beijing, China). The *bamGal4 vp16* line was a gift from Professor Zhaohui Wang at the Institute of Genetics and Developmental Biology, Chinese Academy of Sciences. All the flies were raised in 150 mL conical flasks with standard corn/sugar medium at 25 °C.

### 4.2. Fertility Test

Gene-knockdown flies (*bamGal4 > ValRS-m-hp*) were generated by crossing transgenic RNAi males with virgin *bamGal4* females. Flies from the cross between w1118 males and *bamGal4* females were used as the control (*bamGal4>+*). The 1-day-old gene knockdown males were crossed with 3–4-day-old w1118 females to assay male fertility. For each biological repeat, 15 males and 10 females were used. After 12 h of crossing, the males were removed, and then the eggs were collected and incubated at 25 °C for 24 h. The hatching rates were determined by the proportion of hatched eggs to the total eggs. Three biological repeats per cross type were performed.

### 4.3. Immunofluorescence Staining

One-day-old fly testes were dissected in PBS, fixed for 30 min in 4% paraformaldehyde at room temperature, and washed in wash buffer (phosphate-buffered saline (PBS)/0.1% Triton X-100) three times for 15 min each. The testis samples were blocked for 30 min in 5% normal goat serum and incubated overnight at 4 °C with the primary antibodies. The primary antibodies were used at the following dilutions: rabbit anti-Vasa (1:100, cat. no. AB760351, Developmental Studies Hybridoma Bank, Iowa, IA, USA), mouse anti-spectrin (1:100, cat. no. AB528473, Developmental Studies Hybridoma Bank), and anti-ATP5A. The secondary antibodies were used at the following dilutions: phalloidin (1:200, cat. no. BMD00084, Abbkine, Wuhan, China), rabbit 594, and mouse 488 (1:200, cat. no. A23420 and A23210, Abbkine). All the samples were mounted on glass slides with 4′-6-diamidino-2-phenylindole (DAPI) (2 μg mL^−1^, cat. no. S2110, Solarbio, Beijing, China) solution [59]. Fluorescence images were collected using a Leica SP8 laser confocal microscope (Leica, Wetzlar, Germany).

### 4.4. Transmission Electron Microscopy

The testes were dissected from 1d *bamGal4/ValRS-m-hp* and *bamGal4>+* males in PBS and fixed in 2.5% glutaraldehyde (0.2 M phosphate buffer, pH 7.4) at 4 °C overnight. Then, these testes were washed in phosphate buffer and post-fixed in 1% OsO_4_ for one hour. After double fixation, the samples were dehydrated through an ascending series of ethanol (30%, 50%, 70%, 80%, 100%, ten minutes for each concentration and doubled for 100%) and then embedded in Araldite (EMbed 812, Beijing, China). Ultrathin sections (80 nm) were obtained using a Leica Ultracut 7. The sections were placed on copper grids, stained with 2% uranyl acetate for 15 min, rinsed twice with H_2_O for 5 min, and then stained with lead citrate for 15 min. The stained samples were observed using an H-8100 transmission electron microscope (Hitachi, Tokyo, Japan) operating at 100 kV.

### 4.5. TUNEL Staining

TUNEL staining (TdT-mediated dUTP nick-end labeling) was performed as follows: Thirty pairs of 1-day-old *Drosophila* testis were dissected and fixed in 4% paraformaldehyde solution at RT for 40 min, then washed four times (15 min each time) in PBST (phosphate buffer, PBS + 0.1% Triton X-100). The samples were then incubated with the TUNEL reaction mixture (5 μL TdT enzyme solution and 45 μL fluorescein-dUTP tag solution) in a dark environment at 37 °C for three hours (Roche, Germany). After rinsing three times with PBST in the dark, we used an anti-fading medium (Solarbio, Beijing, China) containing 2 μg/mL DAPI to fix the testes on the glass slides. A Leica SP5 laser confocal microscope was used to observe and take photos.

### 4.6. RNA Extraction, Library Preparation, and Sequencing

To screen the genes related to *ValRS-m* during spermatogenesis, we collected 80–100 testes samples from *bamGal4/ValRS-m-hp* and *bamGal4>+* males. These testes were used for the transcriptomic analysis. Six cDNA libraries were constructed and sequenced on the Illumina Novaseq 6000 platform (San Diego, CA, USA) following the standard protocols set by Majorbio Bio-Pharm Technology Co., Ltd. (Shanghai, China). Subsequently, clean reads were mapped to the genome of *Drosophila* melanogaster (FlyBase, dmelr6.15 genome) using TopHat 2.1.1 software. After assembly of the mapped reads, the unigenes were run against a non-redundant database.

To identify the DEGs (differentially expression genes) between two different groups, the expression level of each gene was calculated according to the transcripts per million reads (TPM) method. RSEM was used to quantify the gene abundances [60]. The DEGs were determined with a cutoff of |log2 fold change (RNAi/control)| ≥ 1 and *p*-value ≤ 0.05, as per the criteria used in a previous study [61]. In addition, functional enrichment analyses, including GO and KEGG, were performed to identify which DEGs were significantly enriched in GO terms and metabolic pathways at a *p* value ≤ 0.05 compared with the whole-transcriptome background. The GO functional enrichment and KEGG pathway analyses were carried out using Goatools and KOBAS [62].

### 4.7. Quantitative Real-Time PCR

The assay was performed as previously described [18]. The total RNA was extracted using TRIzol (cat. no. 15596026, Invitrogen, Waltham, MA, USA) from the testes and ovaries of 1-day-old adult w1118, as well as the testes from bamGal4/ValRS-m-hp and bamGal4>+ males. The first-strand cDNA was synthesized from 2 μg of total RNA using an EasyScript first-strand cDNA synthesis SuperMix Kit (cat. no. AT321-01, TransGen Biotech, Beijing, China). qPCR was performed with specific primers (Appendix A) using a Miniopticon system (Bio-Rad, Hercules, CA, USA) with Platinum SYBR Green qPCR SuperMix (cat. no. Q711-02, Vazyme, Shanghai, China). The qPCR cycling program was as follows: 95 °C for 3 min, followed by 40 cycles of 95 °C for 10 s, 55–60 °C (depending on the different primers) for 30 s and 65 °C for 5 s. Then, a melting curve was constructed from 55 °C to 98 °C. The relative expression of the gene was calibrated against the reference gene *rp49* using the 2^−ΔΔCT^ calculation method: ΔΔCt = (Ct_Target_ − Ct_rp49_)*ValRS-m* RNAi − (Ct_Target_ − Ct_rp49_) control [63].

### 4.8. Statistical Analysis

Statistical analysis was performed using GraphPad Prism 10 software (GraphPad Inc., La Jolla, CA, USA). A student’s *t*-test was used for the two-group comparisons. A value of *p* < 0.05 was considered significantly different. The statistical details are indicated in the corresponding figure legends.

## 5. Conclusions

In conclusion, our study demonstrated that *ValRS-m* is required for normal mitochondrial behavior and meiosis-related gene expression during *Drosophila* spermatogenesis. The most striking finding was that *ValRS-m* acts as a novel regulator in *Drosophila* testes, which governs the transition from spermatogonia into spermatocytes. That is the reason why *ValRS-m*-knockdown impaired mature sperm production and male fertility (Figure 8). Further investigations should determine how *ValRS-m* acts as a switch from spermatogonia to spermatocytes during *Drosophila* spermatogenesis.

## Figures and Tables

**Figure 1 ijms-25-07489-f001:**
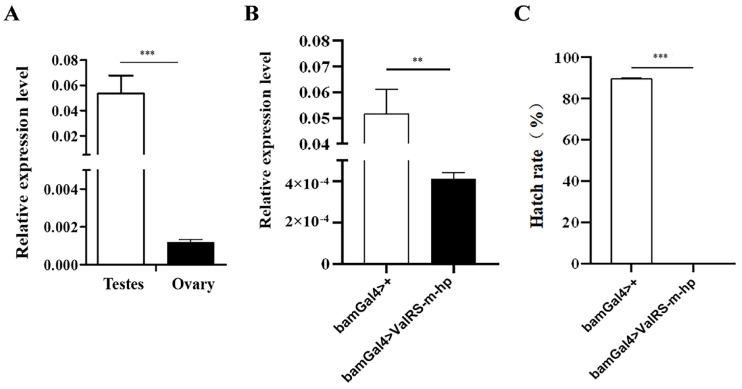
Knockdown of *ValRS-m* in testes caused *Drosophila* male infertility. (**A**) The *ValRS-m* expression level in 1-day-old adult testes was significantly higher than that in the ovaries, as shown by qRT-PCR. (**B**) The *ValRS-m* expression level was significantly reduced in *Bam Gal4 > ValRS-m-hp* (*ValRS-m* RNAi) testes. (**C**) The hatching rate of eggs from the crossing of wild-type females and *ValRS-m* RNAi males was zero. ** *p* < 0.01; *** *p* < 0.001.

**Figure 2 ijms-25-07489-f002:**
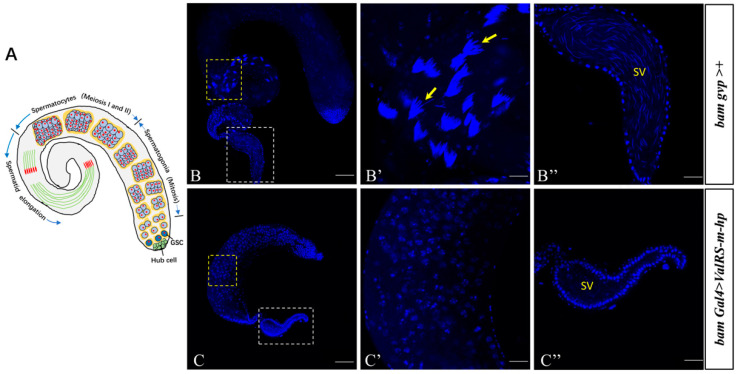
*ValRS-m* is crucial for male fertility. (**A**) Diagram illustrating *Drosophila* spermatogenesis. Immunofluorescent staining using DAPI for spermatogenesis in the control testis (**B**–**B″**) and *ValRS-m*-knockdown testis (**C**–**C″**). (**B**,**C**) are the whole testes. The enlarged view of the dashed boxes in (**B**,**C**) is shown in (**B′**,**B″**,**C′**,**C″**), respectively. (**B′**) shows the spermatid nuclei bundles (yellow arrows) in the control testes. (**B″**,**C″**) are the seminal vesicles (SV). Many mature sperm nuclei are visible in the SV of the control group (**B″**), but no sperm nuclei are visible in the SV of the *ValRS-m*-knockdown testes (**C″**). DNA was stained with DAPI (blue). Scale bars, 100 μm (**B**,**C**); 50 μm (**B′**,**B″**,**C′**,**C″**).

**Figure 3 ijms-25-07489-f003:**
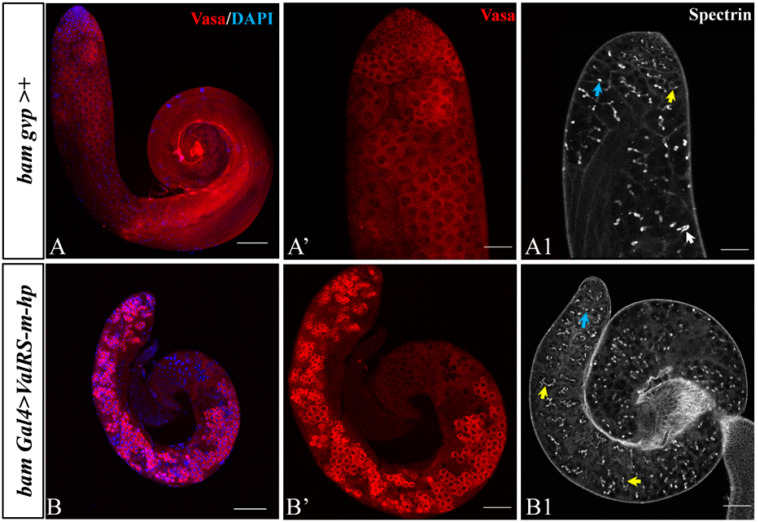
Reduction of *ValRS-m* restrained spermatogonia differentiation. (**A**,**B**) Early germline differentiation in the control testis and *ValRS-m*-knockdown testis. Vasa (red) is a pan-germ-cell marker (**A**,**A′**,**B**,**B′**). Spectrin antibody labels fusomes (**A1**,**B1**). The punctate fusomes (blue arrow) indicate the GSCs and their gonialblast progeny cells, whereas the thin-branched fusomes (yellow arrow) indicate spermatogonia and the thick-branched fusomes (while arrow) indicate spermatocytes. DNA was stained with DAPI (blue). Scale bars: 100 µm (**A**,**B**); 50 µm (**A′**,**B′**,**A1**,**B1**).

**Figure 4 ijms-25-07489-f004:**
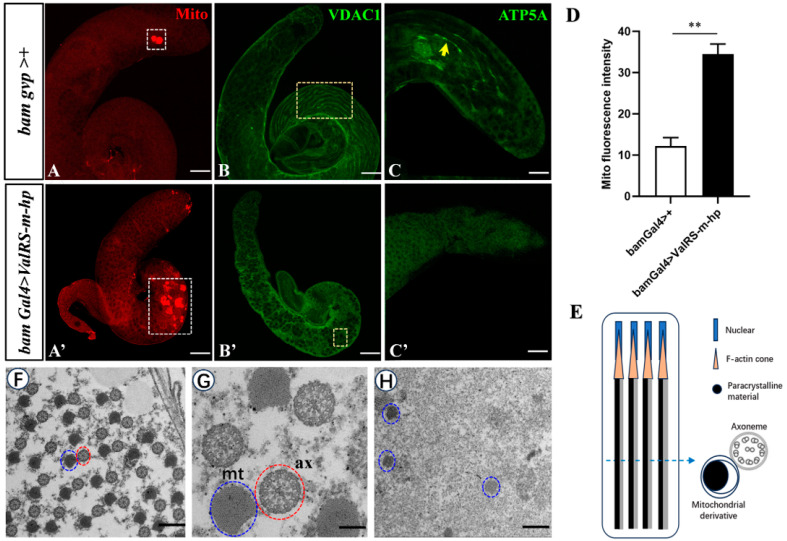
*ValRS-m*-knockdown caused mitochondrial defects in testes. (**A**–**C**): Control testis. (**A′**–**C′**): *ValRS-m*-knockdown testes. The white boxes in (**A**,**A′**) indicate mitochondria by MitoTracker staining. Yellow boxes in (**B**,**B′**) indicate the mitochondrial envelope morphology (VDAC1 markers). The yellow arrows in (**C**) indicate ATP signaling (ATP5A markers). (**D**) The mean mitochondrial staining (Mito) signal (n = 8) detection in testes. (**E**) Schematic diagram depicting the mitochondrial phenotypes of elongating spermatids in control testes. The dashed arrows indicate the location of cross-sections corresponding to the schematic diagrams. (**F**–**H**) Transmission electron microscopy (TEM) images showing mitochondrial morphogenesis. The enlarged view of (**F**) is shown in (**G**). In the representative image of control testes (**F**,**G**), an axoneme (ax, outlined by a red circle) interacts with a mitochondrial derivative (mt) with paracrystalline material (outlined by a blue circle). A few scattered mitochondrial derivatives exist, but any axonemes are not visible in the *ValRS-m* RNAi testes (**H**). ** *p* < 0.01. Scale bars, 100 μm (**A**–**C**,**A′**–**C′**); 1 μm (**F**,**H**); 200 nm (**G**).

**Figure 5 ijms-25-07489-f005:**
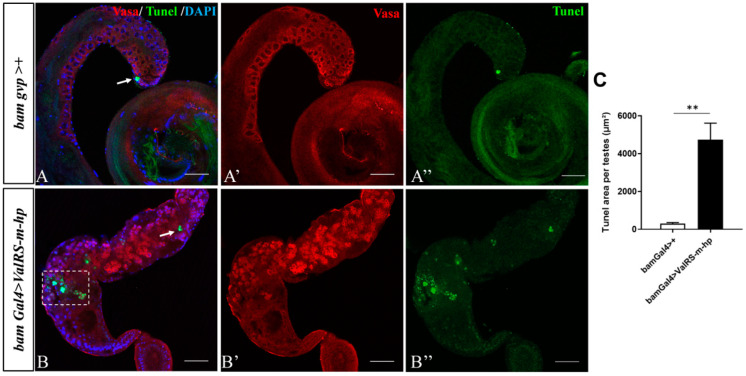
Knockdown of *ValRS-m* induced apoptosis in the testes. DAPI (blue) stained the nuclei. Vasa (red) stained the germ cells. The single channel images for Vasa staining are shown in (**A′**,**B′**); the single channel images for TUNEL staining (green) are shown in (**A″**,**B″**); and the merge signal in the testes is shown in (**A**,**B**). The white arrow and white box indicate the TUNEL signals in (**A**,**B**). (**C**) The area with TUNEL signals in control and *ValRS-m*-knockdown testes (n = 10) was measured and quantified. ** *p* < 0.01. Scale bars: 50 µm.

**Figure 6 ijms-25-07489-f006:**
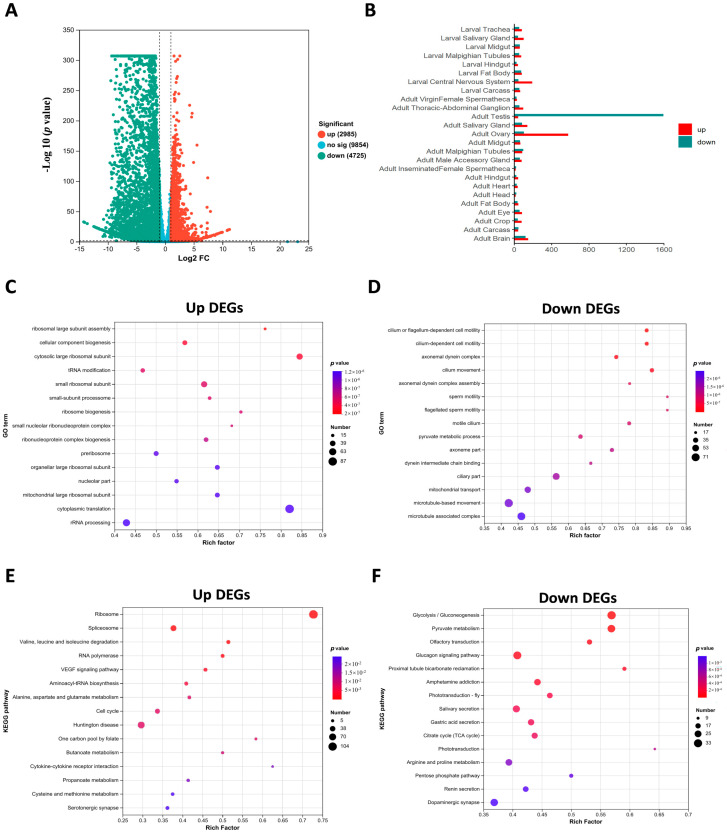
Transcript alterations were assessed using RNA-seq in *ValRS-m*-knockdown testes. (**A**) Volcano plot of differentially expressed genes (DEGs) from the comparison of the control and *ValRS-m*-knockdown groups. (**B**) Tissue expression profile of DEGs in *ValRS-m*-knockdown testis. (**C**) GO enrichment analysis of up-DEGs in the testes of *Bam Gal4 > ValRS-m-hp* relative to the control. (**D**) GO enrichment analysis of down-DEGs in the testes of *Bam Gal4 > ValRS-m-hp* relative to the control. (**E**) KEGG enrichment analysis of up-DEGs in the testes of *ValRS-m* RNAi relative to the control. (**F**) KEGG enrichment analysis of down-DEGs in the testes of *ValRS-m* RNAi relative to the control.

**Figure 7 ijms-25-07489-f007:**
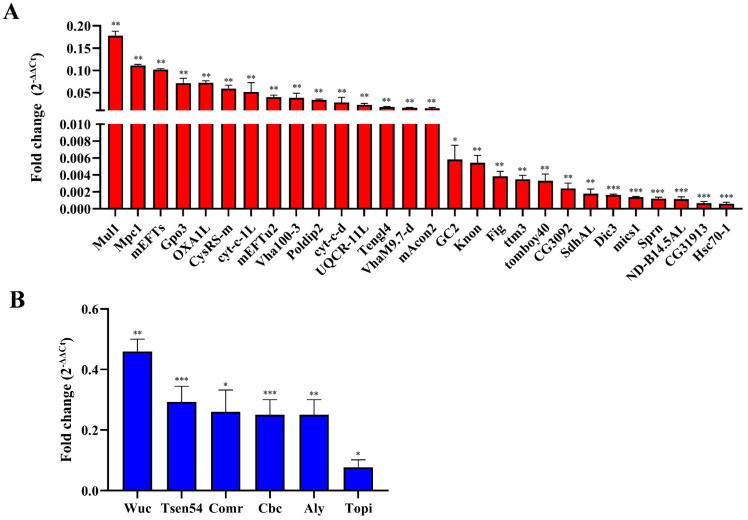
The qRT-PCR validation of DEGs in *ValRS-m*-knockdown testes. (**A**) DEGs associated with mitochondria. (**B**) DEGs related to meiosis. * *p* < 0.05; ** *p* < 0.01; *** *p* < 0.001.

**Figure 8 ijms-25-07489-f008:**
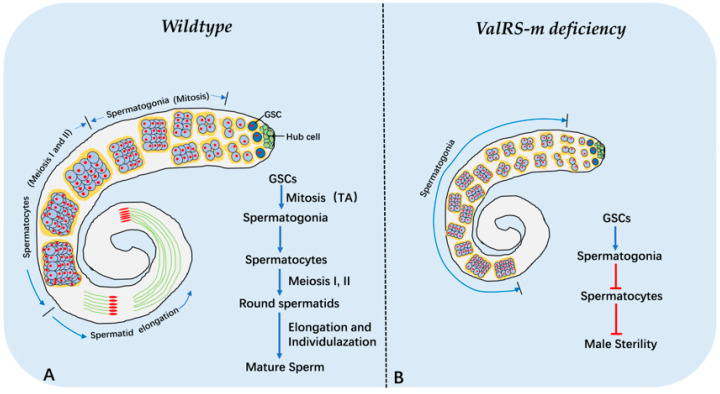
A schematic summary illustrates the phenotypes of wild-type and *ValRS-m*-knockdown in *Drosophila* testes. (**A**) Spermatogenesis in control flies; (**B**) *ValRS-m*-knockdown induce male sterility.

**Table 1 ijms-25-07489-t001:** Classification of DEGs that were down-regulated (fold change ≥ 2, *p* value < 5%) in *ValRS-m*-knockdown males compared with controls via qRT-PCR.

Annotation ID	Gene Name	Description
CG2075	*Aly* (always early)	The onset of spermatid differentiation
CG13493	*Comr* (cookie monster)	Involved in spermatogenesis and transcription regulation
CG8484	*Topi* (matotopetli)	Male meiotic division and spermatid differentiation
CG12442	*Wuc* (Wake-up-call)	Male meiotic nuclear division
CG5970	*Cbc* (crowded by cid)	Required at the transition to meiosis in spermatogenesis
CG5626	*Tsen54* (tRNA splicing endonuclease subunit 54)	Involved in tRNA processing
CG14128	*Sprn* (Spermitin)	Testis-specific mitochondrial lumen protein
CG1287	*mics1* (Mitochondrial morphology and cristae structure 1)	Maintenance of mitochondrial morphology
CG11196	*Dic3* (Dicarboxylate carrier 3)	Mitochondrial dicarboxylate carrier
CG8330	tomboy40	Importation of protein precursors into mitochondria
CG6691	*ttm3* (tiny tim 3)	Mitochondrial protein translocation
CG12201	*GC2* (Glutamate Carrier 2)	Catalyzes the transport of L-glutamate across the inner mitochondrial membrane
CG4706	*mAcon2* (Mitochondrial aconitase 2)	Enables 4 irons, 4 sulfur cluster binding activity
CG30354	*UQCR-11L* (Ubiquinol-cytochrome c reductase 11 kDa subunit-like)	Mitochondrial electron transport
CG13263	*cyt-c-d* (Cytochrome c distal)	Electron carrier protein, sperm individualization
CG12736	*mEFTu2* (mitochondrial translation elongation factor Tu 2)	Brings aminoacyl-tRNA to the ribosome during the elongation phase of mRNA translation
CG14508	*cyt-c1L* (Cytochrome c1-like)	Enables ubiquinol-cytochrome-c reductase activity
CG8257	*CysRS-m* (Cysteinyl-tRNA synthetase, mitochondrial)	Enables ATP binding activity and cysteine-tRNA ligase activity
CG6404	*OXA1L* (OXA1L mitochondrial inner membrane protein)	Mitochondrial proton-transporting ATP synthase complex assembly
CG7311	*Gpo3* (Glycerophosphate oxidase 3)	Enables glycerol-3-phosphate dehydrogenase (quinone) activity
CG6412	*mEFTs* (mitochondrial translation elongation factor Ts)	Recharges the products of mEFTu1 and mEFTu2 with GTP
CG14290	*Mpc1* (mitochondrial pyruvate carrier)	Transports pyruvate across the mitochondrial inner membrane
CG1134	*Mul1* (mitochondrial E3 ubiquitin protein ligase 1)	Involved in the control of mitochondrial morphology by promoting mitochondrial fission
CG6914	*ND-B14.5AL* (NADH dehydrogenase (ubiquinone) B14.5 A subunit-like)	Mitochondrial electron transport, NADH to ubiquinone
CG5718	*SdhAL* (Succinate dehydrogenase, subunit A (flavoprotein)-like)	Mitochondrial respiratory chain complex II
CG7813	*Knon* (knotted onions)	Nebenkern assembly
CG14909	*VhaM9.7-d* (Vacuolar H+ ATPase M9.7 subunit d)	Proton-transporting ATPase activity
CG4683	*Tengl4* (Testis EndoG-Like 4)	Active in mitochondrial inner membrane and nucleus
CG30329	*Vha100-3* (Vacuolar H+ ATPase 100kD subunit 3)	ATPase binding activity
CG12162	*Poldip2* (Polymerase (DNA-directed), delta interacting protein 2)	Active in mitochondrial nucleoid and nucleus
CG7615	*Fig* (fos intronic gene)	Active in mitochondrion
CG8937	*Hsc70-1* (Heat shock protein 70 cognate 1)	ATP hydrolysis activity
CG3092	CG3092	Protein insertion into mitochondrial inner membrane from matrix
CG31913	CG31913	Mitochondrial cytochrome c oxidase assembly

## Data Availability

The raw sequence data are available from the National Center for Biotechnology Information Short Read Archive database (http://www.ncbi.nlm.nih.gov/sra/) (accessed on 1 May 2024) under accession number PRJNA1108791. All other data are available in this text and Appendix A.

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
