# Peer review of "Deficiency of ValRS-m Causes Male Infertility in Drosophila melanogaster"

_ijms, 2024, doi:10.3390/ijms25137489_

Round 1

Reviewer 1 Report

Comments and Suggestions for Authors

In their manuscript entitled "Deficiency of ValRS-m Causes Male Infertility in Drosophila melanogaster", Duan et al. present some interesting findings that ValRS-m modulates spermatocyte differentiation by controlling the levels of several gene transcripts in the Drosophila testis. They found that knockdown of ValRS-m resulted in the accumulation of spermatogonia in almost the whole testis, but no spermatid nuclear bundles appeared at the base of the testis and no mature sperm were present in the seminal vesicle. In particular, they found that knockdown of ValRS-m disrupted mitochondrial fusion and ATP synthesis in fly testes. Finally, they demonstrated a significant down-regulation of meiosis-related genes and several mitochondrial-related genes by ValRS-m RNAi. These results suggest that ValRS-m is essential for Drosophila spermatogenesis by affecting mitochondrial dynamics and function during spermatocyte differentiation. However, some revisions are required before this paper can be accepted.

1. Since ValRS-m is screened from the comparative proteomics of ocn RNAi and control fly testes, is there any evidence showing the relationship between these two genes? Please present.

2. The authors found that ValRS-m functions in male fertility in adult flies, so what about the spatio-temporal expression profiles of this gene? is it specifically highly expressed in adult males and/or testes?

3. The authors describe that ValRS-m is homologous to the human gene VARS2, so what are the functions of VARS2 in humans? This needs to be added in the Introduction or Discussion.

4. The authors found that ValRS-m RNAi caused many gene expression changes. Which of the DEGs were reported to be involved in spermatogenesis or male fertility? These need to be added in the Discussion.

5. Many figures (e.g. Fig. 2A'', B'', A1, B1; Fig. 3A-C) are too small to show the local defect phenotypes. The author should not always show the whole testis. When showing local defects, they should enlarge the local defects so that the reader can clearly see the differences between control and RNAi. 

6.  Some of the English writing seemed to be not professional English. I suggest the authors to revise the whole manuscript.  

Comments on the Quality of English Language

English writing need to be improved before publication.

Reviewer 2 Report

Comments and Suggestions for Authors

This is an interesting study of the genetic regulation of spermatogenesis in Drosophila melanogaster. Authors used different methodological approaches to check their hypothesis. The results obtained seem to be relevant, but authors provide puzzling information on the meiotic process and the differentiating process of germ cells that make difficult to understand the biological impact of their findings. It is frequently difficult to know which germ cell generation they refer to, and which is the main division and/or differentiation process occurring in each.

Moreover, in several sentences authors refer to the meiosis in a wrong way. Note that this divisions requires two cell divisions, the meiosis I (reductional) and meiosis II (equational), to render haploid cells. Sometimes it is not clear if the meiosis is also associated to the differentiation of spermatocytes, and which are the main features of this differentiation process.

Figures also need some improvement. Therefore, some figure legends provide information that is not fully evident from fluorescence images, mainly because the images included are too small and some specific details are difficult to see.

The text contains several grammatical errors that frequently make difficult to understand the message.

Other minor comments are detailed below:

Line 14. Please, refer to “results in” instead of "induced”.

Lines 15-16. Please, refer to ”thereby resulting in the accumulation of spermatogonia in the seminiferous epithelium” instead of “and caused accumulated spermatogonia in testes”.

Line 16. Which “destruction” do the authors refer to? Please, use an appropriate term.

Lines 17-18. “A total of 7710 differentially expressed genes (DEGs) were identified to have at least a 2-fold change with a p-value of <0.05”. Which is the message? Please, revise and improve this sentence.

Lines 21-22. “We also confirmed that six meiosis-related genes expression reduced significantly in ValRS-m RNAi testes”. Please, revise and improve this sentence to provide a clear message.

Lines 22-23. “28 DEGs associated with mitochondria were verified to be suppressed when ValRS-m was insufficient”. Again, this sentence need to be revised and improved.

Lines 23-25. These results suggest that ValRS-m play widely and vital role in the mitochondrial function and male germ cells differentiation during spermatogenesis. This conclusion does not seem directly related with the results obtained. What do the authors mean with “widely and vital role”? How was the mitochondrial function assessed? Please, note that this conclusion is too vague.

Line 35. “In the spermatocyte, each germline stem cells”. This sentence in quite confusing, since spermatocytes are differentiating germ cells instead of stem cells. Please, revise and improve this sentence to provide a clear message.

Line 39. Please, refer to “provide” instead of "create” and to “differentiate” instead of "grow”.

Line 40. “It then undergoes two meiosis”. Please, note that meiosis includes two cell divisions, meiosis I and meiosis II. According to this, authors should revise and improve this sentence to provide a clear message.

Line 42. “eventually becoming haploid mature sperm”. Please, note that sperm cells derived from meiosis I and meiosis II are already haploid, and so the to not “become” haploid. Again, authors should review and improve this sentence to provide a clear message.

Line 47. “spermatogonium must undergo meiosis”. Please, note that spermatogonia divide by mitosis, and spermatocytes by meiosis. According to this, authors should revise and correct this sentence.

Line 52. “could result in a rich spermatocyte”. Do the authors mean in the accumulation of spermatocytes in the seminiferous epithelium? If so, please correct this sentence.

Line 53 “spermatocyte differentiation”. Which “differentiation” do the authors refer to? Please, clarify. Maybe, you should mean “division”.

Line 82. “spermatocyte differentiation”. Again, be sure that the term “differentiation” is correct in the context of this sentence.

Lines 94-96. This sentence is too long and difficult to understand. Please, revise it and provide a clear message.

Line 117. “a large of mature sperm”. What do the authors refer to? Are the authors sure that “large” is an appropriate term in the context of this sentence?

Line 120. “severe defects”. Please, specify these defects, otherwise the message is too ambiguous.

Line 173. “indicated by arrowheads”. The figures do not have arrowheads.

Lines 175-176. “Many mature acicular sperm nuclei are visible in the SV of the control group (D’’)”. I was unable to see these mature acicular sperm in this figure.

Line 183. “twice meiosis”. As previously indicated, meiosis requires two cell divisions (meiosis I and meiosis II) to provide haploid cells. So, this term is uncorrect.

Lines 184-185. “The premeiotic germ cells, including GSCs, GBs, and spermatogonia are small”. Please, provide the cell size, otherwise this sentence is quite ambiguous. Also note that this sentence should be written in past tense.

Lines 185-187. This sentence is too long and quite puzzling. Please, revise it and provide a clear message.

Line 187. “After two rounds of meiosis”. Please, refer to “After meiosis”.

Line 187. “64 sperm cells”. Which is the name of this cells? Are they called spermatids as in mammals? If so, use the appropriate terminology.

Line 200. “synchronizing the behavior of germ cells”. It is not clear if the authors refer to all germ cells generations. Please, clarify.

Line 259. “less ValRS-m causes”. Do the authors mean “less ValRS-m expression? If so, please, clarify.

Line 275. “inhibit the differentiation of germ cells”. It is not clear if the authors refer to all germ cell generation or just spermatids. Please, clarify.

Line 304. “Pre-individualization spermatids”. Which type of spermatids are these? Please, specify.

Line 315. “significantly stronger”. If authors use the term “significantly” they should provide the P value as well as the values of fluorescence intensity.

Line 320. Again, spermatocytes not jus differentiate but also divide by meiosis. So, authors should revise the sentence to provide a clear message.

Line 321. “excessive apoptosis”. How was the apoptosis measure? Which is the threshold to consider that the level of apoptosis is excessive? Please, clarify.

Line 413. “meiotic”. Please, refer to either “meiotic division” or “meiosis”.

Line 415. “works together”. Please, use an appropriate term.

Line 417-419. Please, revise and improve this sentence to provide a clear message. Also note that the sentence is not fully correct from a grammatical point of view.

Lines 420-421. “the differentiation of sperm cells”. Do the authors refer to all sperm cell generation or just to spermatids? Please, clariy.

Lines 419-423. Please, revise and improve this sentence.

Line 449. “disappearance”. Are the authors sure that they “disappear”? Is this term correct?

Line 455. “ValRS-m gene is localized in mitochondria”. Do the authors mean in mitochondrial DNA? If so, please clarify.

Line 461. This sentence is quite confusing considering that somatic cells do not divide by meiosis. Please, revise and provide a clear message.

Lines 468-469. Please refer to “sperm differentiation” instead of “sperm formation”.

Line 472. Please, define “mature spermatocytes”. In which cell stage are they?

Lines 482-483. “the tMAC complex helps to activate meiosis of spermatocytes”. How? Please, specify.

Line 538. “hence destroy the spermatogenesis”. Authors should use another term instead of “destroy”.

Lines 630-631. Please, revise this sentence and provide a clear message.

Lines 633-634. “determining whether spermatocytes can differentiate smoothly into sperm clusters and mature sperm”. This sentence is unclear, considering that spermatocytes divide by meiosis thereby producing spermatids which differentiated into mature sperm. Please, revise and improve it.

Comments on the Quality of English Language

English grammar needs extensive revision and improvement.

Reviewer 3 Report

Comments and Suggestions for Authors

the English must be corrected, the major problem is with the images and figure legends, the reader does not know what the images show, in Fifg.1 where is the expression shown , I see only Dapi staining, I do not see any cells and structures described in the legend, the same for the Fig 2  there re no yellow boxes, where are the sperm clusters??? Also  I do not see fusome,  in Fig. 3 again there are no yellow boxes, mitochondrial morphology is not visible how you can tell from TEM that there is a synchronous development????

You must make the images bigger to show what is there and each figure  must be accompanied by the schematic drawings showing what exactly is on the each image, right now all these image are uninformative

Comments on the Quality of English Language

mustbe improved

Reviewer 4 Report

Comments and Suggestions for Authors

Duan et al. examined the role of ValRS-m in Drosophila male germ cell development by knocking down it’s expression through RNAi, and found that this VARS ortholog played critical roles in spermatocyte differentiation in the testes. The reduction of ValRS-m expression led to male specific infertility, as shown by cross progeny inviability and morphology and DAPI stain of the testes. Using immunofluorescence, the authors then showed that when ValRS-m was knocked down, there were a defect in germ cell differentiation and an increase in GSC-like cysts. The authors further examined mitochondrial morphology and ATP synthesis and discovered that defects in both processes existed when ValRS-m was knocked down. Additionally, a high level of apoptosis was observed in the mutant testes, providing further evidence for the important roles that ValRS-m play in male spermatogenesis.  Lastly, the authors performed transcriptomic profiling and identified many genes being differentially regulated when ValRS-m level was decreased, including those with known functions in meiosis.

Overall, the authors provided thorough and well designed experimental evidence to support their conclusion and delineate the potential pathway through which ValRS-m functions in spermatogenesis. The content is well organized and the manuscript it relatively easy to follow. However, a high level of grammatical errors is found throughout the manuscript, and I believe when they are corrected, it will significantly enhance the quality of this manuscript. Almost every sentence has something that doesn’t read smoothly, even though I was able to get the meaning. With the large number of grammar issues, it’s not practical to list them out here. I suggest the authors to carefully go over the manuscript and may also consider professional writing assistance if available.

Major point:

1.        Please make your abstract and the last paragraph of section 1 consistent. Your current abstract doesn’t contain the information about mitochondria and ATP synthesis, while the last paragraph of section 1 doesn’t have the conclusion about the transcriptional analysis.

2.        Figure 1 D’’ and E”: It would be helpful to include a zoomed-in image of the SV region, as the current version doesn’t show much difference as you described in the manuscript.

Comments on the Quality of English Language

please see my comments above. Extensive editing of the manuscript in terms of writing is required to improve its writing quality and make it suitable for publication.

Round 2

Reviewer 2 Report

Comments and Suggestions for Authors

Authors introduced all the modifications required by the reviewers, and so the overall quality of the manuscript has been improved. Nevertheless, the revision version of this manuscript still contains few confusing sentences on the different steps of spermatogenesis. For instance, in line 45 the sentence “spermatogonia must undergo meiosis at the right time” is incorrect, since spermatogonia are unable to divide by meiosis. Therefore, authors should revise the whole document to ensure that a proper information is given.

Comments on the Quality of English Language

English grammar has been improved, but minor grammatical errors are still detected in the text.

Author Response

Comments 1: Authors introduced all the modifications required by the reviewers, and so the overall quality of the manuscript has been improved. Nevertheless, the revision version of this manuscript still contains few confusing sentences on the different steps of spermatogenesis. For instance, in line 45 the sentence “spermatogonia must undergo meiosis at the right time” is incorrect, since spermatogonia are unable to divide by meiosis. Therefore, authors should revise the whole document to ensure that a proper information is given.

Response 1: We are very sorry about this mistake. Thank you for pointing this out. We have checked and revised the description about the spermatogenesis process to ensure that proper information is given. (Line 45, Line 194)

Comments 2: English grammar has been improved, but minor grammatical errors are still detected in the text.

Response 2: We have revised the English grammar in the text on the Quillbot website. (Line 14, Line 20, Line 22, etc.)

Reviewer 3 Report

Comments and Suggestions for Authors

Abstract and Introduction:

The depletion of ValRS-m blocked the mitochondrial behavior

WHAT ISMITOCHONDRIAL BEHAVIOUR???

the ribosomal metabolic pathway

WHAT METABOLIC PATHWAY THE RIBOSOMES HAVE????

when ValRS-m was insufficient.

WRONG WORD

, and spermatocyte-to-sperm morphological transformation

SPERMATOCYTE DOES NOT CHANGE TO SPERM, THE SPERMTID DOES

mitochondria cluster near the nucleus and present a spherical structure

WRONG GRAMMAR

mitochondrial recombination

WHAT IS THAT???? HOW MITOCHONDRLA DNA RECOMBINES???

the number and appearance of mitochondria are similarly to those of  somatic cells dispersed in the cytoplasm.

HOW DO YOU KNOW MITOCHONRIAL APPEARNCE AT LIGHT MICROSCOPY? ALSI WRONG GRAMMAR

in which the contents of major derivatives are relatively dense, while the volume of minor derivatives decreases and there is no accumulation of the contents

WHAT IS THIS CONTENT???? THIS IS NOT A SCIENTIFIC DESCRIPTION

was first discovered as a mitochondrial dynamic gene

WHAT IS THE MEANING OF THIS??? WHAT ARE THE DYNMIC GENES????

, which is located on the mitochondria and highly expressed in the testis.

IS IT LOCATED ON THE SURFACE OGF MITOCHONDRIA????

of spermatogonia differentiation towards to spermatocytes

GRAMMAR

Results and Figure legends

Hatch rate

GRAMMAR, SHOULD BE THE HATCHING

We found that the size of ValRS-m RNAi testes became smaller than that of the control testes

GARMMAR

Finally, plenty of mature sperm

THIS IS NOT A SCIENTIFIC TERM

(A)  A model of Drosophila melanogaster spermatogenesis.

B’ the spermatid nuclei bundles

WHAT IS THAT???

During the germ cell lineage,

 WHAT DO YOU MEAN???

the mitochondrial content in the testes by ValRS-m knocking down was significantly increased, and most of the locations were concentrated at the posterior segment of the testes, indicating a significant increase in cell proliferation

WHAT ARE THE LOCATIONS??? HOW THE LOCATIONS CAN BE CONCENTRATED??

Fig.4 why there are only two dots of mitochondrial staining????

YOU CAN NOT ASSESS MITOCHONDRIAL ENVELOPE MORPHOLOGY BY LIGHT MICROSCOPY STAINING AND ESPECIALLY BY IMMUNOSTAING

BAD QUALITYOF TEM MITOCHONDRIA IMAGE, THE WHOLE FIGURE EXTREMELY CONFUSING

YOU CAN NOT STUDY THE MITOCHONDRIAL MORPHOGENESIS BY TEM.

WHY IS ONLY ONE APOPTOTICC CELL IN THE WHOLE TESTES? AND AT THE TESTES SURFACE???

Comments on the Quality of English Language

BAD GRAMMAR

Reviewer 4 Report

Comments and Suggestions for Authors

The authors have adequately addressed all of my comments.

Author Response

Thank you very much again for your constructive comments.

Round 3

Reviewer 3 Report

Comments and Suggestions for Authors

the manuscript was much improved

Comments on the Quality of English Language

minor grammar errors